# The Intestinal and Skin Microbiome in Patients with Atopic Dermatitis and Their Influence on the Course of the Disease: A Literature Review

**DOI:** 10.3390/healthcare11050766

**Published:** 2023-03-06

**Authors:** Małgorzata Mazur, Hanna Tomczak, Martha Łodyga, Katarzyna Plagens-Rotman, Piotr Merks, Magdalena Czarnecka-Operacz

**Affiliations:** 1College of Health, Beauty Care and Education in Poznań, 60-133 Poznań, Poland; 2Central Microbiological Laboratory, H. Święcicki Clinical Hospital at the Poznan University of Medical Sciences, 60-366 Poznan, Poland; 3Department of Medicine Berkshire Medical Center, Pittsfield, MA 01201, USA; 4Center for Pediatric, Adolescent Gynecology and Sexology Division of Gynecology, Department of Perinatology and Gynecology, Poznan University of Medical Sciences, 61-758 Poznan, Poland; 5Department of Pharmacology and Clinical Pharmacology, Faculty of Medicine, Collegium Medicum, Cardinal Stefan Wyszyński University, 01-938 Warszawa, Poland; 6Allergic and Occupational Skin Diseases Unit, Department of Dermatology, Medical University of Poznań, 60-355 Poznan, Poland

**Keywords:** microbiome, atopic dermatitis

## Abstract

Bacteria inhabiting the digestive tract are responsible for our health. The microbiome is essential for the development of the immune system and homeostasis of the body. Maintaining homeostasis is very important, but also extremely complicated. The gut microbiome is related to the skin microbiome. It can therefore be assumed that changes in the microbes inhabiting the skin are greatly influenced by the bacteria living in the intestines. Changes in the composition and function of microbes (dysbiosis in the skin and intestines) have recently been linked to changes in the immune response and the development of skin diseases, including atopic dermatitis (AD). This review was compiled by collaborating Dermatologists specializing in atopic dermatitis and psoriasis. A comprehensive review of the current literature was performed using PubMed and limited to relevant case reports and original papers on the skin microbiome in atopic dermatitis. The inclusion criterion was that the paper was published in a peer-reviewed journal in the last 10 years (2012–2022). No limitations on the language of the publication or the type of study were made. It has been shown that any rapid changes in the composition of the microflora may be associated with the appearance of clinical signs and symptoms of the disease. Various studies have proven that the microbiome of many systems (including the intestines) may have a significant impact on the development of the inflammatory process within the skin in the course of AD. It has been shown that an early interaction between the microbiome and immune system may result in a noticeable delay in the onset of atopic diseases. It seems to be of high importance for physicians to understand the role of the microbiome in AD, not only from the pathophysiological standpoint but also in terms of the complex treatment that is required. Perhaps young children diagnosed with AD present specific characteristics of the intestinal microflora. This might be related to the early introduction of antibiotics and dietary manipulations in breastfeeding mothers in the early childhood of AD patients. It is most likely related to the abuse of antibiotics from the first days of life.

## 1. Introduction

Atopic dermatitis (AD) is a chronic, recurrent inflammatory disease with characteristic skin lesions and severe pruritis [1]. The disease most often begins in early childhood. It is usually the first manifestation of the so-called allergic march, in which other atopic diseases are also involved (e.g., asthma, allergic rhinitis and allergic conjunctivitis) [2]. The etiopathogenesis of AD is very complex and is related to genetic and environmental factors that induce an abnormal course of the immune system response associated with the predominance of CD4 lymphocyte differentiation toward the Th2 lineage. The consequences of the increased activity of Th2 lymphocytes are the increased production of cytokines released by these cells, especially IL-4, IL-5 and IL-13, and the concomitantly decreased synthesis of IFN-(gamma). People suffering from AD are predisposed to IgE-dependent hypersensitivity to antigens (both extrinsic and intrinsic) [3,4,5]. 

Th22 cells secreting IL-22 [6] play an essential role in the initiation and acute phase of AD, while a transition to Th1/Th17 responses characterizes chronic disease [7].

IL-33, TSLP and type 2 cytokines can function as itch-inducing factors. The dependence between epithelial barrier dysfunction, type 2 immunity and itch is presented in Figure 1.

The role of the skin barrier in the development of AD is also extremely important [8]. Its damage allows the penetration of allergens and colonization of the skin by pathogenic microorganisms. This stimulates the development of inflammation associated with an excessive Th2 lymphocyte response and promotes the breakdown of the skin’s protective barrier, but it also affects distant sites, e.g., in the intestine or the respiratory tract. The early protection of the skin barrier is extremely important because it limits the development of local and general inflammation [9].

Recently, we have encountered a new definition of atopic dermatitis—understood as a chronic inflammatory skin disease characterized by a disturbance of the skin barrier, inflammation and dysbiosis (imbalance between commensal and pathogenic bacteria, which may have significant health implications) [10,11]. Skin dysbiosis is based upon an increased concentration of *Staphylococcus aureus*, which leads to a reduction in the number of commensal bacteria [12,13]. The altered microbiome in AD affects the immune system, stimulating inflammatory reactions manifested by atopic eczema. Knowledge on this subject is important in influencing new strategies for AD treatment and prevention [14].

**Figure 1 healthcare-11-00766-f001:**
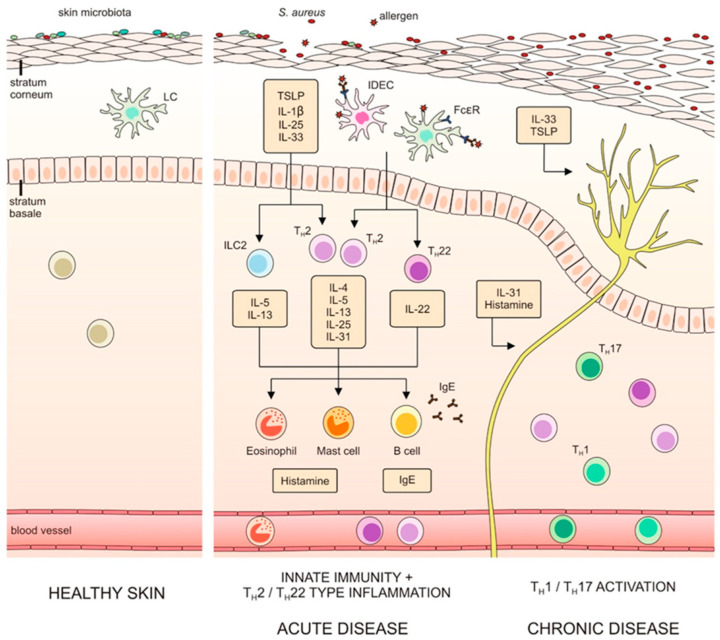
Dependence between epithelial barrier dysfunction, type 2 immunity and itch (with the authors’ consent) [15].

The objective of this study was to determine the influence of the gut microbiome on AD, considering patient age. 

## 2. Materials and Methods

This review was compiled by collaborating Dermatologists specializing in atopic dermatitis and psoriasis. A comprehensive review of the current literature was performed using PubMed and limited to relevant case reports and original papers on the skin microbiome in atopic dermatitis using search terms and MeSH terms such as “microbiome”, “atopic dermatitis”, “neonatal”, “prebiotics” and “probiotics”. The inclusion criterion was that the paper was published in a peer-reviewed journal in the last 13 years (2010–2023). No limitations on the language of the publication or the type of study were made.

## 3. The Skin Microbiome in Atopic Dermatitis

As is known, the human skin microbiome varies depending on many factors, including the body region, age and gender [16,17]. Beneficial microorganisms inhabiting the skin prevent it from being colonized by pathogenic microorganisms. The commensal microflora of the skin protects humans from pathogens and helps keep the immune system in balance between effective protection and destructive inflammation [18]. Commensal microflora presents a highly protective effect: e.g., *Staphylococcus epidermidis* produces antimicrobial substances that fight pathogens, and *Cuticabacterium acnes* uses skin lipids to produce short-chain fatty acids that suppress microbiological hazards [19]. *Cutibacterium* and *Corynebacterium* also reduce the number of *Staphylococcus aureus* bacteria, which are important in terms of influencing the severity of AD. It seems to be highly important to monitor the profile of microorganisms inhabiting the skin of patients with AD. In most cases, as has been previously shown, *S. aureus* is cultured from these individuals. It is the initial colonization that, in the exacerbation phase of the disease, may lead to an infection of the affected skin region. It is not known whether the patients themselves transfer these bacteria to the affected areas with their hands, because the skin itches, or whether deficiencies in immunity due to allergies lead to spontaneous colonization with this pathogen. Meylan et al. presented the results of a study of 149 infants in the first 2 years of life with the aim to analyze whether the concentration of *S. aureus* on the skin is increased in patients before the AD diagnosis. In addition, *Staphylococcus hominis* is less frequently seen in patients who develop AD later in life. Strains of *S. aureus* isolated from AD patients secrete various destructive exotoxins (e.g., staphylococcal toxins can act as superantigens with strong immunostimulating properties, induce specific IgE synthesis and damage the skin barrier) [20].

Additionally, the effect of treatment on *S. aureus* colonization in the course of AD was investigated. Ultraviolet phototherapy has been shown to reduce *S. aureus* colonization in atopic skin and to reduce the production of toxins by these bacteria. Exposure to ultraviolet radiation also induces the production of cathelicidin AMP in the skin, which additionally protects against the effects of *S. aureus* [21]. Kong et al. confirmed that the skin microflora becomes more diverse after AD treatment, which may be related to the maintenance of the remission of skin lesions [22]. A review of published literature supports that probiotic preparations containing *Bifidobacterium* spp. reduce pruritis, which may be related to the fact that these bacteria, when regularly supplied to the body through probiotic supplementation, colonize the skin and, on a competitive basis, minimize the multiplication of *S. aureus* and other pathogenic microorganisms. As is already well known, the skin microbiome is influenced by the area around the body. Examples include different conditions in seborrheic areas, which are inhabited by lipophilic species of *Cutibacterium acnes* (formerly *Propionibacterium*), or in moist areas, which contain abundant *Corynebacterium* and *Staphylococcus spp*. Fungi such as *Malassezia* spp. are abundant on the trunk and arms [23]. It has been shown that the proportions of the presence of two dominant *Malassezia* species on the skin vary depending on the severity of AD. Patients with mild (to moderate) lesions show an advantage of *Malassezia resticta* over *Malassezia globosa*, while the ratio is close to 1:1 in patients with severe AD. In addition, the skin of patients with AD presents a wider variety of yeasts (e.g., *Cryptococcus liquefaciens*, *Candida albicans*, *Cryptococcus diffluens*) compared to healthy individuals [24]. It is of high importance to realize that the skin condition of AD patients can also be influenced by clothing. Windproof fabrics promote the multiplication of anaerobic bacteria, so it is important to recommend to parents that their children use clothes made of natural, breathable materials (e.g., cotton, linen) in order to prevent relapses of the disease.

## 4. The Gut Microbiome in Atopic Dermatitis in Children

The role of the gut microbiome has recently been widely studied in the course of atopic dermatitis (AD). The composition of the gastrointestinal microflora differs in different periods of human life. The appearance of *atopic dermatitis* in young children may therefore be related to problems already present at that stage in shaping the microbiome. This is where an important aspect of age may play a role in AD. Most often, this problem concerns children. Many studies have already confirmed the beneficial effects of *Bifidobacterium* and *Bacteroidetes* found in the intestines of infants and their lower concentrations in adults. These bacteria have a protective function, preventing the intestines from being colonized by pathogens. It has been shown that young children, although carriers of *Clostridioides difficile*, rarely have *Clostridioides difficile* infections (CDIs) the way that adults do. Based upon this observation, one can argue that there may be a connection to the presence of *Bifidobacterium* and *Bacteroidetes*, or their deficiency may have an effect on the body in a way that will contribute to the symptoms of AD [25,26].

### The Gut Microbiome in the Neonatal and Early Childhood Periods

The neonatal digestive tract is initially sterile, but it is soon colonized right after the birth of the baby. The route of delivery and the lack of contact with the mother’s genital tract microflora play an important role in this process [27,28]. A different microflora occurs in babies born by vaginal delivery compared to when the newborn is born via cesarean section. After natural childbirth, bacteria from the mother’s perineum belonging to *Lactobacillus*, *Prevotella* and *Atopobium* are detected. Certain health-promoting species of *Bifidobacterium* and *Bacteroides* are exceptionally abundant in these infants, reducing the regulation of inflammatory responses. On the other hand, after cesarean section, *Staphylococcus*, *Streptococcus* and *Propionibacterium* bacteria are dominant and come from the skin of the mother and/or medical personnel [1,29,30]. These are the first to colonize the skin of newborns. In the first year of life, the intestinal microflora is relatively small, poorly differentiated and subject to change. As bacteria age, the number and diversity of bacteria increase, but the composition of the microbiome becomes stable. Microbes colonize the neonatal intestines at birth and evolve in species abundance until the infant reaches the age of 2–3 years, when the microflora becomes stable [31].

In addition to the method of delivery, the diet in the first moments of a child’s life (breastfeeding vs. artificial feeding) also has a huge impact on the intestinal microbiome. Therefore, the nutrition of newborns, infants and young children should be analyzed. As is known, breast milk has a great influence on the immunity of the baby. It has been noticed that diet plays a large role in the formation of microflora and is of great importance to health [32,33]. Immediately after birth, the intestines of breastfed babies are colonized by *Bifidobacterium* and *Lactobacillus* spp. [34,35]. By implementing a proper diet, we can manipulate the composition of the intestinal microbiota [36]. Bacterial colonization of the gut during early childhood (the first 3 years) has a strong influence on the immune system, which affects the possible development and course of AD. It is necessary to estimate the influence of natural and artificial feeding on the onset of AD signs and symptoms. 

The intestinal microflora of infants has been analyzed by various researchers in order to understand its influence on the developing immune system and, if necessary, the risk of developing allergic diseases. Lee et al. analyzed the composition of the gut microbiome in 129 infants (at 6 months of age) depending on the type of feeding. AD was diagnosed in 63 infants. This study showed that the gut microbiome of children differed depending on the type of feeding. In addition, in children with AD, it was shown that the number of bacterial cells in the stool was lower than in the group of healthy infants, which could be related to antibiotic therapy that destroys the microbiome. Additionally, the analysis of the entire metagenome revealed differences in functional genes related to the development of the immune state, which could be related to the severity of AD [37]. In a recent study by Park et al., the composition and functions of the intestinal microflora and short-chain fatty acids (SCFAs) in 6-month-old infants and their impact on the natural development of AD up to 24 months of age were analyzed. The study of stool samples showed that children with transient AD have low levels of *Streptococcus* bacteria and large amounts of *Akkermansia*. In the case of chronic AD, low levels of *Clostridium* and *Akkermansia* were observed, while the *Streptococcus* level was high. Additionally, the relative number of *Streptococcus* positively correlated with the SCORAD score (Scoring Atopic Dermatitis—assessing the severity of atopic dermatitis), while the number of *Clostridiums* correlated negatively. The study showed that the composition and functions of the intestinal microbiome may be related to the course of AD [38]. In an analysis of the intestinal microbiota in infants depending on the feeding method, it was shown that formula-fed infants have higher concentrations of *Escherichia coli* (*E. coli*) and *Clostridium* (ex. *C. difficile*) [39]. This observation may be due in part to the fact that artificial milk has added sugars. The introduction of solid food causes a dynamic shift in the intestinal microflora from the dominant line of *Bifidobacterium* to *Bacteroides* and *Clostridium*. In a study by Watanabe et al., it was shown that the number of *Bifidobacterium* in AD patients was significantly lower than in healthy subjects. Moreover, the number of *Bifidobacterium* and their percentage varied depending on the severity of the disease state (fewer bacteria were found in people with severe AD). On the contrary, *Staphylococci* were observed more frequently in AD patients than in healthy individuals [40]. Fieten et al. identified the so-called “bacterial signature”: the fecal microbiota had relatively more *Bifidobacterium pseudocatenulatum* (*B. pseudocatenulatum*) and *E. coli* and fewer *B. adolescentis*, *B. breve*, *F. prausnitzii* and *Akkermansia muciniphila* (*A. muciniphila)* [41].

Lee et al. confirmed that intestinal bacterial colonization in infancy is closely related to the development of the immune system, and abnormal intestinal microflora precedes the onset of atopic disease [42]. Some studies have shown that infants with AD have low intestinal bacterial differentiation, apart from small amounts of *Bifidobacterium* and *Bacteroides* and high levels of *Enterobacteriaceae* [43]. Lee et al. demonstrated that in patients with AD, the proportions of *Clostridia* (including *Clostridium difficile*), *E. coli* and *S. aureus* in the intestinal microbiome are higher than in healthy controls [44].

Interestingly, the findings suggest that there is an interaction between the gut microflora and the skin microbiome [45]. *S. aureus* is the most common pathogen cultured from the skin of AD patients, and exacerbations of the disease are associated with superantigens of these bacteria. However, a recent cohort study found that intestinal colonization by *S. aureus* strains was negatively associated with the later development of AD in infancy. Such strains can stimulate and promote the infant’s immune maturation. Although *S. aureus* of the skin can exacerbate AD, this does not exclude the possibility that prior mucosal colonization of the intestines by *S. aureus* prior to the “atopic march” may be protective (through immune stimulation provided by this species). Indirectly, the gut microbiome can modulate blood cytokine levels, thereby affecting brain function, anxiety levels and stress. Cortisol is released under stressful conditions and can alter the permeability of the intestinal epithelium, thus modifying its barrier function and subsequently altering the composition of the intestinal microbiome [46]. It also alters the levels of circulating neuroendocrine molecules such as tryptamine, trimethylamine and serotonin, thereby transforming the skin barrier and skin inflammatory response. These neuroendocrine molecules can be considered as future AD drugs [47]. Sugar content is also very important in the diet in terms of the multiplication of certain groups of microorganisms. In diabetics, infections with etiologies of *S. aureus* and *Candida* spp. are very common. With too much sugar in the diet, *E. coli* multiplies in the intestines and causes serious changes in the microbiome. It is possible that the addition of sugar to artificial milk causes the “bad” bacteria to multiply. Alternatively, a proper diet, rich in vegetables and fruits, contributes to the growth of beneficial bacteria belonging to the *Bacteroidetes* group. One can contemplate whether children with AD have deficiencies or inadequate concentrations of these bacteria. 

Studies have proven that antibiotic therapy also destroys the microbiome of both the intestines and the skin and may lead to dysbiosis [48,49]. It is also worth considering the other general changes that antibiotics cause in the microbiome, both in the intestines and in the skin. These drugs cannot fight only germs at the site of infection. After antibiotic treatment, the microbiome rebuilds itself within about 2 years, but in terms of diversity, it will never return to normal [50,51]. The question at hand is, did these AD-related changes originate from antibiotic therapy in early childhood? Often, antibiotics are used repeatedly and over a long period of time. It is possible that initial changes in the skin, incorrectly diagnosed as an allergy, were treated with local antibiotics, which led to disturbances in the microbiota and, as a consequence, serious changes and colonization by pathogens. Patients with AD also often suffer from respiratory problems, and antibiotics are often prescribed when a respiratory infection is suspected with simple symptoms such as coughing (and then secondary infections). It is important to collect accurate data on the length of elapsed time since the last day of antibiotic use when submitting material for research on the microbiome in AD. Only then will the data accurately reflect the real situation. Topical steroids also facilitate colonization with pathogens, and these drugs are used from the very first symptom onset in children.

## 5. Prebiotics, Probiotics and Symbiotics

The microbiome has a huge impact on health and, therefore, sometimes needs to be modified. Over the past 50 years, it has been suggested that manipulating the gut microbiome could become an important form of treatment in pathologic processes. It is also observed that an important role in disease is played not only by pathogenic bacteria but also by the deficiency or absence of commensal microorganisms [52,53]. The microbiome can be manipulated through the use of probiotics. Their task is to have an immunomodulatory effect [52]. However, it should be noted that probiotics, whether they are applied topically or taken orally, colonize the intestines or skin only during the treatment period. Studies show that after the cessation of therapy, probiotic strains are detected only for a short period of time [54]. Connections between influencing factors in the microbiome in patients with AD are becoming more frequently studied. Even the effectiveness of including prebiotics or probiotics in AD treatment regimens is being analyzed by researchers. 

### 5.1. Probiotics

Probiotics are live bacteria and yeast cultures that are considered to be beneficial for the general well-being of humans. Orally administered probiotics provide benefits by interacting with the gut microflora, while when applied topically, they act by modulating the skin microflora [55]. Prebiotics, on the other hand, contain inanimate fibers that stimulate the growth and activity of beneficial microorganisms. Popular probiotic families are the Gram-positive bacteria *Bifidobacterium* and *Lactobacillus*. Probiotics modulate the immune system by stimulating the differentiation of regulatory T cells and the production of anti-inflammatory cytokines (TGF-β and IL-10). In addition, *Lactobacillus accelerates* the reconstruction of the skin barrier and inhibits skin inflammation associated with the substance. *Bifidobacterium bacteria* have an antipruritic effect [56,57]. Unfortunately, the effects of using probiotics in human studies have shown conflicting results. Rautava et al. analyzed the effect of probiotic supplementation in pregnant women and nursing mothers on reducing the risk of developing eczemic lesions in children. *Lactobacillus rhamnosus* GG *(L. rhamnosus* GG) taken by pregnant women prevented the development of atopic dermatitis in half of the children from the high-risk group up to 2 years of age, with stabilization up to 4 years of age. Infants who were given *Bifidobacterium lactis (B. lactis) Bb-12* or *L. rhamnosus* strain GG developed milder forms of AD [58]. 

Most of the studies confirm that supplementation with single-strain preparations during pregnancy (mainly *Lactobacillus rhamnosus* GG) prevents AD in high-risk infants [59,60,61,62].

In 2015, the World Allergy Organization (WAO) published guidelines on the use of certain probiotic strains during pregnancy. WAO experts recommend using probiotics in pregnant and breastfeeding women whose children and infants are at high risk of developing AD [63,64]. It should be emphasized that the WAO recommendations are in compliance with the guidelines of other scientific societies, e.g., the European Society for Paediatric, Gastroenterology, Hepatology and Nutrition (ESPGHAN) [65]. 

Clinical studies by Kalliomäki et al. [66] showed that the administration of the probiotic *Lactobacillus rhamnosus* strain GG (ATCC 53103) in the third trimester of pregnancy and continuation of supplementation for six months in breastfeeding mothers or infants (if mix-fed) reduced the incidence by half in AD in children aged 2, 4 and 7 years. 

Nermes et al. [61] investigated the interaction of *Lactobacillus rhamnosus* GG (LGG) with the skin and gut microbiota and humoral immunity in infants with AD. The proportions of IgA- and IgM-secreting cells decreased significantly in the treated vs. untreated group and at 1 month were 0.59 (95% CI 0.36–0.99, *p* = 0.044) for IgA- and 0.53 (95% CI 0.29–0.96, *p* = 0.036) for IgM-secreting cells. The proportions of CD19 + CD27 + B cells increased in the probiotic-treated infants, but not in the untreated ones. On the skin, the bacterial counts of *Bifidobacterium genus* vs. *Clostridium coccoides* in treated and untreated infants were similar. Probiotics reinforce the intestinal barrier function and support the immune response [61]. 

The benefits of probiotics were also observed in studies by Woo [60], with improvements of at least 30% and 50%. 

The issue of atopy treatment using probiotics is more debatable. The current state of knowledge indicates that single-strain preparations are not always effective, as opposed to multi-strain probiotics and prebiotic preparations. A randomized double-blind placebo-controlled study carried out in Poland among 60 children with an allergy to cow milk manifested as atopic dermatitis showed the significant effectiveness of probiotic strains (*Lactobacillus casei* ŁOCK 0900, *L. casei* ŁOCK 0908 and *L. paracasei* ŁOCK 0919) in improving the condition of the patients by more than half [67]. 

A study by Gueniche et al. assessed the effect of *Lactobacillus paracasei* supplementation on skin sensitivity and transepidermal water loss (TEWL). The study observed a reduction in skin sensitivity and transepidermal water loss in healthy adults using supplementation [68]. The authors attributed these effects to an increase in circulating TGF-β because this cytokine was shown to affect skin barrier integrity. The study confirmed the concept of the interdependence of the skin and gut in terms of the interaction of the microbiome. However, their beneficial effects have been questioned [69]. According to the analysis, probiotics did not cause significant differences in skin symptoms as assessed by participants or parents. There was also no evidence that probiotics had an effect on the quality of life (QoL) in patients with eczemic lesions. The probiotics slightly reduced the severity of skin lesions as assessed by the investigator, but this was not clinically sufficient. Therefore, this study questioned the use of probiotics as an element of the complex treatment of eczema. Peterson et al. analyzed the literature concerning studies conducted on the assessment of the intestinal microflora in patients with AD. The results of the above-mentioned analyses were unfortunately inconclusive. Almost half of the included intervention studies showed that altered intestinal bacterial colonization caused by the use of probiotics presented a positive effect on the severity of AD. However, other studies did not confirm this effect. The researchers emphasized that the role of the gut microbiome requires further research, as it still remains controversial [70].

### 5.2. Prebiotics

Prebiotics enhance the production of acetate, propionate and butyrate, which have anti-inflammatory effects, reduce the generation of toxic fermentation products, increase lymphocyte and/or leucocyte numbers in gut-associated lymphoid tissues and increase intestinal IgA secretion [71].

### 5.3. Symbiotics

Symbiotics are synergistic combinations of probiotics and prebiotics, intended to positively influence the condition of the skin and intestines.

They affect the development of useful intestinal microflora by stimulating probiotics with prebiotics and by hindering the growth of pathogenic gut flora. Symbiotics reduce concentrations of undesirable metabolites in the body, superoxide nitrosamines and cancerogenic substances, act against putrefaction in the gut and prevent constipation and diarrhea. Studies among rats whose diet contained inulin, oligofructose, *Lactobacillus rhamnosus* and *Bifidobacterium lactis* showed higher intestinal IgA levels.

Symbiotics also reduce noxious microflora (*Clostridium perfringens* and other endopathogens), promote the proliferation of useful bacteria [72] and improve the absorption of calcium, magnesium and phosphorus [73]. 

Based on the reviewed literature data, there is a strong suggestion that differences in the composition and ratio of the gut microbiome are associated with the generation of many neurotransmitters and neuromodulators. These highly active substances may further influence the clinical course of AD by affecting the skin barrier (both structure and function) as well as by having an important impact on regulatory aspects of the immune system. However, due to inconclusive data, additional studies are necessary to confirm the real role of intestinal dysbiosis in AD. Perhaps special attention should be directed toward the time from which the last dose of a probiotic or antibiotic was administered to when samples (stool or skin swab) are collected for analysis. In order to appropriately identify the real importance and role of the gut or cutaneous microbiome in AD development and the course of the disease, both of them should be analyzed in detail in terms of their different pathways of action and bilateral interactions, as well as the final influence of the regulatory function of the immune system in general. Additionally, it seems that specific probiotic strains, even though no longer isolated, do not introduce significant changes in the entire intestinal and skin microbiome. It is most probable that the introduced strains facilitate the maintenance of the adequate regulation of the immune system, while other species do not dominate. Due to the fact that, in AD, the intestinal microbiota seems to be, in a specific way, an important factor in the process of development and in the course of AD, intestinal microbiota transplants from a healthy donor (FMT) have been proposed as a new therapeutical option. This option requires the complete replacement of intestinal microorganisms, along with the entire environment of the AD recipient, with the microflora obtained from the feces of a healthy donor [74,75]. FMT is known to be an effective method for the permanent exchange of bacteria with their entire environment. It is a method especially helpful in a wide spectrum of diseases related to intestinal dysbiosis [76]. Although there are still contradictory opinions concerning the role of probiotics in the prevention and course of AD, including their place as a possible therapeutical option, the position of probiotics as an alternative treatment strategy in the upcoming “post-antibiotic” era seems to be of high interest and importance. 

Promoting breastfeeding, encouraging vitamin D supplementation in pregnant women and infants, and restricting antibiotics early in life may reduce the risk of AD. In the near future, drugs and supplements containing personalized microbial strains may play an important role in the treatment of AD as well [77].

## 6. Conclusions

In order to appropriately identify the real importance and role of the gut or cutaneous microbiome in the development and course of AD, both of them should be analyzed in detail in terms of their different pathways of action and bilateral interactions, as well as the final influence of the regulatory function of the immune system in general.Although there are still contradictory opinions concerning the role of probiotics in the prevention and course of AD, including their place as a possible therapeutic option, the position of probiotics as an alternative treatment strategy in the upcoming “post-antibiotic” era seems to be of high interest and importance.

## Data Availability

Not applicable.

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
