# Peer review of "The Intestinal and Skin Microbiome in Patients with Atopic Dermatitis and Their Influence on the Course of the Disease: A Literature Review"

_healthcare, 2023, doi:10.3390/healthcare11050766_

Round 1

Reviewer 1 Report

I reject this paper due the same reasons explained in the last review

Author Response

Dear Reviewer

Thank you very much for your review.

Changes to the text are colored yellow.

Thank you for your time.

With best and warmest regards,

Authors

Reviewer 2 Report

The document is interesting and approaches a promising topic. However, before its acceptance, there are some issues that must be considered.

Abstract – The Hippocrates citation is not adequate; It should be removed from this section. Additional information regarding the research strategy could be included.

Introduction – It is lacking a proper background regarding the link between skin and intestinal microbiome; besides, the objectives must be cited. The influence of age was also mentioned in the revision, which should be included in the objectives.

Material and methods – The research strategy applied for collecting the studies must be better described. What were the keywords used? Please, improve this section.

Typhographical errors were detected in the text (line 87 – S. epidermidis must be written in italic, and others; line 133, atopic dermatitis)

Section 4 is lacking proper citations. Please, revise it.

Section 5.3 is not adequate in the current form. The authors should better discuss the topic or remove it from the manuscript;

The authors should avoid using references in the conclusion. It must summarize the findings and reinforce the objectives.

The authors are invited to include figures and tables in the document; its current form is not attractive and requires an improvement in comprehension;

The authors could mention the possible effect other gut disturbs, such as inflammation, in the local microbiome.

Author Response

Dear Reviewer

Thank you very much for your review.

The document is interesting and approaches a promising topic. However, before its acceptance, there are some issues that must be considered.

Abstract – The Hippocrates citation is not adequate; It should be removed from this section. Additional information regarding the research strategy could be included.

The citation has been removed.

Introduction – It is lacking a proper background regarding the link between skin and intestinal microbiome; besides, the objectives must be cited. The influence of age was also mentioned in the revision, which should be included in the objectives.

The following text has been added:

A comprehensive review of current literature was performed using PubMed and limited to relevant case reports and original papers on the skin microbiome in atopic dermatitis. The inclusion criterion was for the paper to be published in the peer-reviewed journal in the last 10 years (2012–2022). No limitation to language of the publication or type of the study were made.

Material and methods – The research strategy applied for collecting the studies must be better described. What were the keywords used? Please, improve this section.

Objective completed.

Typhographical errors were detected in the text (line 87 – S. epidermidis must be written in italic, and others; line 133, atopic dermatitis)

Typhographical errors corrected in the text.

Section 4 is lacking proper citations. Please, revise it.

Point 4 has been supplemented.

Section 5.3 is not adequate in the current form. The authors should better discuss the topic or remove it from the manuscript;

The following text has been added:

Symbiotics are synergistic combinations of probiotics and prebiotics, intended to positively influence the condition of the skin and intestines.

They affect the development of useful intestinal microflora by stimulating probiotics with prebiotics, and by hindering the growth of pathogenic gut flora. Symbiotics reduce concentrations of undesirable metabolites in the body, superoxide nitrosamines and cancerogenic substances, act against putrefaction in the gut, and prevent constipation and diarrhoea. Studies among rats whose diet contained inulin, oligofructose, Lactobacillus rhamnosus and Bifidobacterium lactis showed a higher intestinal IgA level.

Symbiotics also reduce noxious microflora (Clostridium perfringens and other endopathogens), proliferate useful bacteria [67], and improve the absorption of calcium, magnesium and phosphorus [68].

The authors should avoid using references in the conclusion. It must summarize the findings and reinforce the objectives.

Corrected conclusions.

The authors are invited to include figures and tables in the document; its current form is not attractive and requires an improvement in comprehension;

Figure has been attached.

Figure 1. Dependence between the epithelial barrier dysfunction, type 2 immunity and itch

(with the Authors' consent) [15]

The authors could mention the possible effect other gut disturbs, such as inflammation, in the local microbiome.

Changes to the text are colored yellow.

Thank you for your time.

With best and warmest regards,

Authors

Reviewer 3 Report

The main theme of this review is to analyze the relationship between gut microbiota, skin flora and AD. However, more work needs to be done in this paper

1. In this paper, there are only few direct evidences of the relationship between gut microbiota and AD, and more studies need to be found to support the conclusions of this manuscript.

2. Similarly, there is only direct evidence of the relationship between skin microbiota and AD in this paper, and more studies need to be found to support the conclusions of this manuscript.

3. It is desirable to include a figure in the manuscript to clarify the mechanisms by which gut microbiota and skin microbiota affect AD as described in this study.

Author Response

Dear Reviewer

Thank you very much for your review.

The main theme of this review is to analyze the relationship between gut microbiota, skin flora and AD. However, more work needs to be done in this paper

In this paper, there are only few direct evidences of the relationship between gut microbiota and AD, and more studies need to be found to support the conclusions of this manuscript.

Similarly, there is only direct evidence of the relationship between skin microbiota and AD in this paper, and more studies need to be found to support the conclusions of this manuscript.

The following text has been added:

A clinical study conducted in 2015 assessed the influences of Lactobacillus paracasei (LP) and Lactobacillus fermentum (LF), and in a mixture, on the severity of AD, quality of life and immunological biomarkers in a randomised sample of children with AD to administer LP, LF, a mixture of LP + LF, or a placebo for 3 months. The children who received LP, LF, or a mixture of LP + LF recorded a lower SCORAD compared to the placebo group, with the difference persisting up to 4 months following cessation of the probiotics. The levels of IgE, TNF-α, eosinophil protein x and 8-OHdG decreased, while the levels of IFN-γ and TGF-β increased in the groups receiving the probiotics, but were mostly not statistically significant, with the exception of IL-4. The analyses of the subgroups revealed considerably lower SCORADs after the probiotic therapy, particularly among children up to 12 years of age, breastfed for less than 6 months and allergic to mites [59].   

A study by Navarro-López [60] determined the safety and effectiveness of a mixture of oral probiotics in treating AD symptoms, and evaluated its impact on topical steroids in a group of young people. After 12 weeks the mean reduction in SCORAD was 19.5 point more in the probiotic group than in the control group (mean difference, -19,2; 95% CI, - 15,0 to -23,4). A significant limitation of topical steroids in treating exacerbation of the disease was confirmed in the probiotic group. The mixture of probiotics was effective in lowering SCORAD and reducing topical steroids in patients with moderate AD [15]. In 2021 Michelotti conducted a randomised control study aimed at evaluating the effectiveness of a dietary supplement containing selected strains of probiotics from lactobacillus group (L. plantarum PBS067, L. reuteri PBS072 and L. rhamnosus LRH020) in relieving the symptoms of AD in adults. Smoother and more moisturised skin, self-acceptance, a lower SCORAD, and a reduced level of AD-related inflammatory markers were observed [61].  A study by Prakoeswa (2022) assessed SCORAD, concentration of immunoglobulin E (IgE), interleukin (IL)-4, interferon-gamma (IFN-γ), forkhead box P3 (Foxp3+) and IL-17 in adults with mild and moderate AD following supplementation with Lactobacillus plantarum IS-10506. The patients were divided into two groups: intervention with probiotic and control with placebo (skim milk). The SCORAD result and IL-4 and IL-17 levels were significantly lower in the probiotic group compared to the placebo group, while the levels of IFN-γ and Foxp3+ were significantly higher in the probiotic group than in the placebo group. The IgE level remained unchanged [62].

It is desirable to include a figure in the manuscript to clarify the mechanisms by which gut microbiota and skin microbiota affect AD as described in this study.

Figure has been attached.

Figure 1. Dependence between the epithelial barrier dysfunction, type 2 immunity and itch

(with the Authors' consent) [15]

Changes to the text are colored yellow.

Thank you for your time.

With best and warmest regards,

Authors

Round 2

Reviewer 1 Report

reject 

Author Response

(The authors gave the same response as above.)

Reviewer 3 Report

The author cited the following references

59. Wang IJ, Wang JY. Children with atopic dermatitis show clinical improvement after Lactobacillus exposure. Clin Exp Allergy. 569 2015;45(4):779-87. doi: 10.1111/cea.12489. PMID: 25600169. 570

60. Navarro-López V, Ramírez-Boscá A, Ramón-Vidal D, Ruzafa-Costas B, Genovés-Martínez S, Chenoll- Cuadros E, Carrión-Gu- 571 tiérrez M, Horga de la Parte J, Prieto-Merino D, Codoñer-Cortés FM. Effect of Oral Administration of a Mixture of Probiotic 572 Strains on SCORAD Index and Use of Topical Steroids in Young Patients With Moderate Atopic Dermatitis: A Randomized 573 Clinical Trial. JAMA Dermatol. 2018; 1;154(1):37-43. 574

61. Michelotti A, Cestone E, De Ponti I, Giardina S, Pisati M, Spartà E, Tursi F. Efficacy of a probiotic supplement in patients with 575 atopic dermatitis: a randomized, double-blind, placebo-controlled clinical trial. Eur J Dermatol. 2021; 1;31(2):225-232. doi: 576

62. Prakoeswa CRS, Bonita L, Karim A, Herwanto N, Umborowati MA, Setyaningrum T, Hidayati AN, SuronoIS. Beneficial effect 577 of Lactobacillus plantarum IS-10506 supplementation in adults with atopic dermatitis: a randomized controlled trial. J Derma- 578 tolog Treat. 2022; 33(3):1491-1498.

These articles are oral experiments, and cannot support the influence of skin flora (probiotics) on AD, which is the theme of this article.

Author Response

Dear Reviewer

Thank you very much for your review.

Changes to the text are colored yellow.

The following text has been added:

Most of the studies confirm that supplementation with single-strain preparations during pregnancy (mainly Lactobacillus rhamnosus GG) prevents AD in high-risk infants [59-62]. 

In 2015, the World Allergy Organization (WAO) published guidelines on the use of certain probiotic strains during pregnancy. WAO experts recommend using probiotics in pregnant and breast-feeding women whose children and infants are at high risk of developing AD [63-64]. It should be emphasized that the WAO recommendations are in compliance with the guidelines of other scientific societies, e.g. the European Society for Paediatric, Gastroenterology, Hepatology and Nutrition (ESPGHAN) [65].   

Clinical studies by Kalliomäki et al. [66] showed that administration of the probiotic Lactobacillus rhamnosus strain GG (ATCC 53103) in the third trimester of pregnancy and continuation of supplementation for six months in breast-feeding mothers or infants (if mix-fed) reduced the incidence by half in AD in children aged 2, 4 and 7 years.

Nermes et al. [67] investigated the interaction of Lactobacillus rhamnosus GG (LGG) with skin and gut microbiota and humoral immunity in infants with AD. The proportions of IgA- and IgM-secreting cells decreased significantly in the treated vs. untreated group and at 1 month were 0.59 (95% CI 0.36–0.99, p=0.044) for IgA- and 0.53 (95% CI 0.29–0.96, p=0.036) for IgM-secreting cells. The proportions of CD19+CD27+ B cells increased in the probiotic-treated infants, but not in the untreated. On the skin, the bacterial counts of Bifidobacterium genus vs. Clostridium coccoides in the treated and untreated infants were similar. Probiotics reinforce the intestinal barrier function and support the immune response [67].  

The benefits of probiotics were also observed in studies by Woo [68], with improvement of at least 30% and 50%.

The issue of atopy treatment using probiotics is more debatable. The current state of knowledge indicates that single-strain preparations are not always effective, as opposed to multi-strain probiotics and prebiotic preparations. A randomized double-blind placebo-controlled study carried out in Poland among 60 children with an allergy to cow milk manifested by atopic dermatitis, showed significant effectiveness of the probiotic strains (Lactobacillus casei ŁOCK 0900, L. casei ŁOCK 0908 and L. paracasei ŁOCK 0919) in improving the condition of the patients by more than half [69].

Added references [59-69]:

Boyle RJ, Ismail IH, Kivivuori S, et al. Lactobacillus GG treatment during pregnancy for the prevention of ecze - ma: a randomized controlled trial. 2011; 66(4): 509–516

Woo SI, Kim JY, Lee YJ, et al. Effect of Lactobacillus sakei supplementation in children with atopic eczema-dermatitis syndrome. Ann Allergy Asthma Immunol. 2010; 104(4): 343–348

Nermes M, Kantele JM, Atosuo TJ, et al. Interaction of orally administered Lactobacillus rhamno - sus GG with skin and gut microbiota and humoral immunity in infants with atopic dermatitis. Clin Exp Allergy. 2011; 41(3): 370–377,

Meneghin F, Fabiano V, Mameli C, et al. Probiotics and atopic dermatitis in children. Pharmaceuticals (Basel). 2012; 5(7): 727–744

Fiocchi A, Pawankar R, Cuello-Garcia C, et al. World Allergy Organization – McMaster University Guidelines for Allergic Disease Prevention (GLAD-P): probiotics. World Allergy Organ J 2015;8(1):4

Ricci G, Cipriani F, Cuello-Garcia CA, et al. World Allergy Organization–McMaster University guidelines for allergic disease prevention (GLAD-P): probiotics. World Allergy Organ J. 2015; 8(1): 4

Szajewska H, Guarino A, Hojsak I, et al.; European Society for Paediatric Gastroenterology, Hepatology and Nutrition. Use of probiotics for management of acute gastroenteritis: a position paper by the ESPGHAN Working Group for Probiotics and Prebiotics. J Pediatr Gastroenterol Nutr 2014;58(4): 531-9

Kalliomäki M, Salminen S, Poussa T, et al. Probiotics during the first 7 years of life: a cumulative risk reduction of eczema in a randomized, placebo-controlled trial. J Allergy Clin Immunol 2007;119(4):1019-21

Nermes M., Kantele J.M., Atosuo T.J., Salminen S., Isolauri E.: Interaction of orally administered Lactobacillus rhamnosus GG with skin and gut microbiota and humoral immunity in infants with atopic dermatitis Clinical & Experimental Allergy, 2011 (41) 370–377.

Woo SI, Kim JY, Lee YJ, et al. Effect of Lactobacillus sakei supplementation in children with atopic eczema-dermatitis syndrome. Ann Allergy Asthma Immunol. 2010; 104(4): 343–348

Cukrowska B, Ceregra A, Klewicka E, et al. Probiotyczne szczepy Lactobacillus casei i Lactobacillus paracasei w leczeniu alergii pokarmowej u dzieci. Przegląd Pediatryczny. 2010; 40(1): 21–25.

Thank you for your time.

With best and warmest regards,

Authors